# Knowledge, Attitude, and Practice of Adolescent Parents on Free Sugar and Influencing Factors about Recognition

**DOI:** 10.3390/ijerph17114003

**Published:** 2020-06-04

**Authors:** Qiong Tang, Qian Lin, Qiping Yang, Minghui Sun, Hanmei Liu, Lina Yang

**Affiliations:** Department of Nutrition Science and Food Hygiene, Xiangya School of Public Health, Central South University, 110 Xiangya Road, Changsha 410078, China; 176914092@csu.edu.cn (Q.T.); linqian@csu.edu.cn (Q.L.); yangqiping12@csu.edu.cn (Q.Y.); sun.1234@csu.edu.cn (M.S.); hanmeiliu@csu.edu.cn (H.L.)

**Keywords:** free sugar, knowledge, attitude, practice, factors

## Abstract

Physiological problems caused by excessive intake of free sugar have been concerning important public health issues, especially the impact on adolescents. The World Health Organization (WHO) strongly recommends controlling daily sugar intake in order to reduce the stress caused by high sugar uptake. Finding out the factors that affect adolescents’ sugar intake will help further interventions to control the intake of sugar. Therefore, we conducted a cross-sectional study among 10 middle schools in Changsha, the capital of Hunan province. Two classes of the first and second grades were randomly selected from each school, and their parents in these classes participated in the survey. Questionnaires were used to access the status of adolescent parents’ knowledge, attitude and practice (KAP) towards free sugar. Out of 1136 valid participants, 70.4% of respondents were female with the mean (Standard Deviation) age of 41.76 (±5.27) years. They had a good attitude but relatively poor knowledge and behavior towards free sugar. Binary logistic regression analysis found that parents whose gender is female, with a high education level and a girl as their child, hold a high level of free sugar recognition. These findings could help free sugar control interventions for adolescents in the future.

## 1. Introduction

Free sugar includes monosaccharides and disaccharides added to food and beverages by manufacturers, cook or consumers, as well as sugars naturally present in honey, syrups, fruit juices and fruit juice concentrates. Nevertheless, the endogenous sugars in whole fresh fruits and vegetables do not belong to this group. Added sugar is also a widely accepted concept, and its main difference from free sugar is that free sugar contains natural sugars in nonwhole fruits and vegetables. That is, added sugar is part of free sugar [1,2]. Free sugars are simple sugars, and studies have shown that carbohydrates are the main source of energy for the body. Contrary to complex carbohydrates derived from intact and unprocessed vegetables, excessive amounts of simple carbohydrates can easily lead to health problems such as weight gain, while complex carbohydrates can prevent diet-related chronic noncommunicable diseases [3]. Epidemiological and clinical studies reported that added sugar was associated with the prevalence of obesity and diabetes. Sugar-sweetened beverages (SSBs), one of the main sources of free sugar intake, have also been confirmed to result in overweight/obesity, type 2 diabetes and cardiovascular disease and other unhealthy health consequences [4,5,6]. Sugar intake in adolescents is still at a high level [7,8,9], the main sources being SSBs, biscuits, candies, cakes, breads and so on [10,11,12].

From 2002 to 2012, the consumption rate of sugary food for residents over the age of two in China showed an upward trend, from 20.04% to 26.90%. The overall consumption trend was higher in cities than in rural areas, and higher in women than in men. Children and adolescents were high-consumption groups, and the sugary food consumption rate among adolescents aged 12–18 increased from 21.4% to 27.4% (consumption rate—the proportion of people who consumed sugary foods one or more times during the three days of the survey) [13]. In 2015, 10.7% of sugar intake of Chinese children and adolescents exceeded 25 g/d, and sugar intake from beverages accounted for the largest proportion of prepackaged foods, with an average of 13.2 g/d [14]. In recent years, developed countries have taken many strategies to reduce sugar intake, but sugar intake is still a problem needing to be addressed. The medium daily sugar intake of adolescents aged 9–18 in the United States accounts for 14.1% of total energy, and the total sugar intake of children and adolescents in Europe accounts for 11–17% [15,16]. The ELANS study conducted in Latin American found that there were 78.4% of adolescents whose energy intake from added sugar exceeded 10% of total daily energy. Furthermore, 70% of Swedish adolescents consumed free sugar exceeding 10% of total energy [17,18].

Studies have showed that free sugar is an important risk factor for the occurrence of noncommunicable diseases. A high-sugar diet means excessive calorie intake, which not only increased the risk of obesity, diabetes, dental caries, and nonalcoholic fatty liver disease, but also caused metabolic changes such as lipid derangements, coronary heart disease, and other vascular diseases [19,20,21].

Adolescent sugar intake is closely related to parents and family factors. Studies showed that adolescents’ sugar intake was positively correlated with parental intake, and adolescents consumed much more sugar at home (65%) than they consumed outside (35%) [22,23]. Mothers’ sugary food expenditure also affected children’s free sugar consumption. For example, children of mothers who consumed soft drinks every day were 1.8 times more likely to take foods with free sugars as those of mothers who never consumed soft drinks [24]. A study on the dietary relationship between Norwegian adolescents and parents found that the intake of adolescents’ sugary drinks was significantly positively related to parental intake, and was significantly negatively related to parental education level [25]. In addition, a study conducted in China found that adolescents with low parental education level were more likely to drink SSBs than others [26].

Therefore, this study intends to investigate the knowledge, attitudes and practices of adolescents’ parents about free sugar, understand the current situation of adolescents’ parents buying/storing behaviors of beverages and desserts, and behaviors of sugar-containing food in adolescents, in order to lay the foundation for further control of adolescents’ free sugar intake in the future.

## 2. Materials and Methods 

### 2.1. Study Design 

Data analyzed by the cross-sectional study was collected among parents whose children were studying at middle school between May to July 2019 in Changsha, Hunan. It is a medium-sized city and has certain representativeness nationwide. With the assistance of the Changsha Education Bureau and district education bureaus, the survey team used stratified sampling to sample in five districts in Changsha. In the first stage, two middle schools were randomly selected from each of the five administrative districts, and a total of 10 schools were included. In the second stage, two classes were randomly selected from the first grade and the second grade of each school. A total of 40 classes were surveyed. This study consists of a parental questionnaire and student physical measurements. The questionnaire needed to be filled out by the primary caregivers. The students’ physical measurements and information collection are surveyed and filled in by investigators who have been uniformly trained. The survey was advanced after approval by the Education Bureau, also was approved by the Ethics Committee of Xiangya School of Public Health, Central South University (IRB number: XYGW-2019-025). 

### 2.2. Participants

We calculated the required sample size through the formula to be 824. Considering the 20% no response rate, the minimum survey sample was finally determined to be 989. We collected 1136 subjects to meet the minimum requirements. All students in the class were the survey object, except for students or parents who did not agree to participate. Written informed consent was obtained from all participants before the survey.

### 2.3. Questionnaire Investigation

A structured questionnaire was completed by consulting relevant domestic, foreign literature and related materials and after many rounds of expert discussion. The questionnaire contains four parts.

#### 2.3.1. Demographic Information

This demographic data included parents’ gender, age, ethnicity, height, weight, education level and monthly household income per capita, and were collected through the first part of the questionnaire. When taking physical measurements such as height and weight of students, their age and gender were also collected.

#### 2.3.2. Parent’s Knowledge of Free Sugar

The knowledge section consists of 4 single-choice questions and 2 multiple-choice questions and was aimed at assessing and evaluating parents’ awareness of the definition of free sugar and free sugar content of food, sugar intake guidelines, and the perception of the effects of free sugars on health. For this section, each option could be correctly selected to score 1 point, and incorrectly selected (or uncertain) to score 0 points. The total score was 12 points.

#### 2.3.3. Parent’s Attitude to Free Sugar

The Attitude section contains 6 entries about parents’ attitudes towards eating too much sugary food, drinking too many sugary drinks can be addictive and reducing/controlling children’s intake of sugary foods (free sugar). For the attitude section, selecting “strongly agree” and “agree” scored 1 point, other options were not counted, and the highest score was 6 points. Higher scores indicate a more positive attitude, and parents were more supportive of controlling youth sugar/free sugar intake.

#### 2.3.4. Parental Practice of Free Sugar

The behavior part includes the behavior of parents buying or storing beverages and desserts for their children and boot behaviors of parents in the previous month before the survey. There were 8 items in total. For the practice section, a score of 1 was given for choosing the answer reflecting a good practice and 0 was given for choosing the answer reflecting poor practice. The total score was 8 points.

#### 2.3.5. Classification of Knowledge, Attitude and Practice

We divided them into two categories according to the 75th percentile of the respondents’ knowledge, attitude, and practice scores. Values greater than P_75_ were positioned as high levels of knowledge, positive attitudes, and good behaviors. Otherwise, they were defined as low levels of free sugar knowledge, negative attitude and poor performance. A score of ≥9 indicates good cognition, ≥6 indicates positive attitude, and ≥7 indicates good behavior.

### 2.4. Data Processing and Statistical Analysis

Data entry was performed using EpiData 3.1 software (The EpiData Association, Odense, Denmark), with double entry and consistency check. Statistical analysis was performed using SPSS 18.0 software (IBM Corp., Armonk, NY, USA). Counting data were expressed as mean ± standard deviation (Mean ± SD), and measurement data were expressed as percentages. Demographic data and KAP scores were descriptive statistical methods. After statistical testing, knowledge, attitude and behavior scores followed a skewed distribution, and the difference in scores between groups was analyzed using a nonparametric test. Multivariate analysis was performed using binary logistic regression analysis. Spearman correlation analysis was used to describe the strength of the relationship between knowledge, attitude and behavior. *p* < 0.05 indicated statistical significance.

## 3. Results

### 3.1. Sociodemographic Data

We recovered a total of 1212 questionnaires, and after removing incomplete questionnaires, there were 1136 left, with an effective recovery rate of 93.7%. There were 336 male parents (29.6%) and 800 female parents (70.4%), with an average age of 41.76 ± 5.266 years. Most parents were Han, only 3.6% of parents were ethnic minorities; 69.8% of them were mothers, 28.6% were fathers, 1.6% were grandparents. More than half (57.5%) of families had only one child. Parents who had received university education or above account for 30.5% of the total number, and there were still 2.3% of parents had only primary education, as shown in Table 1.

### 3.2. Knowledge, Attitude and Practice (KAP) Related to Free Sugar

The average scores of knowledge, attitude, and behavior were 7.04, 5.17 and 5.39. The proportion of people with scores above the 75th percentile was 21.2% (knowledge), 51.1% (attitude), and 19.2% (practice). Nearly half of the parents had a positive attitude, but parents’ knowledge and practice of free sugar were not very good (Table 2).

#### 3.2.1. Free Sugar-Related Knowledge

Nearly one-fifth of the parents had a high level of free sugar awareness. The lowest score of knowledge item was for the questions ‘Know the definition of free sugar’. According to Table 3, 7.4% of parents reported they knew “free sugar”. Furthermore, 34.0% of parents knew the daily sugar intake recommended in “The Chinese Dietary Guidelines”, and 30.5% of parents could choose the correct daily sugar restriction recommended by the World Health Organization. About 80% of the respondents could correctly distinguish the free sugar content in biscuits, cakes, vegetables, and fruits, but only 27.5% of parents could correctly recognize that the free sugar content in 100% fruit juice was higher. Most parents (90.1%) agreed that excessive intake of sugar-sweetened beverages will affect adolescent growth and development. The results showed that 75.4% of parents recognized that excessive sugar intake is related to dental caries, and 89.9% of parents recognized that excessive sugar intake is related to overweight/obesity, but parents had low awareness of chronic diseases (41.7%) in adulthood (Table 3).

#### 3.2.2. Free Sugar-Related Attitude

Table 4 showed the answers of parents’ attitudes to free sugar. Only 63.1% of parents thought it is not good to eat too much sugary food. However, 80.1% of parents agreed that drinking too much SSBs would be addictive; and more than 90% of parents agreed to reduce the consumption of sugary foods by the teenage years, limit the amount of sugar added to drinks and desserts, limiting sugary foods is good to children’s health, and parents should consciously control children’s intake of sugary foods.

#### 3.2.3. Free Sugar-Related Practice

Table 5 shows the performance of parents buying and storing beverages, desserts, and boot behaviors one month before the survey. Only a few parents bought more than three kinds of beverages (5.9%) or desserts (8.5%) for their children. More parents stored desserts (60.6%) than stored drinks at home (36.7%). A total of 36.7% of parents stored drinks at home, and 60.6% of parents stored desserts. More than 90% of parents did not use drinks or desserts as a reward for their child’s performance. However, those who consciously avoided buying sugary drinks or drinking sugary drinks in front of children were 34.7% and 37.1%.

#### 3.2.4. Correlation between Knowledge Scores and Practice Scores

After relevant analysis, there was a positive correlation between knowledge scores and behavior scores of parents (R^2^ = 0.019, *p* < 0.000, Figure 1).

### 3.3. Relevant Factors Affecting the Knowledge of Adolescent Parents

We took parents’ gender, age, education level, family’s monthly income per capita, kinship with children, parents’ correctness of children’s body size, student gender, age, classification of students’ body type, and whether they are only children as independent variables, and parents’ high/low level of free sugar awareness as the dependent variable for binary logistic regression analysis.

The results showed that parents’ gender, parents’ education level and students’ gender had effects on parents’ free sugar cognition. We found that women’s awareness of free sugar was higher than men. As education level rises, parents had better level of free sugar awareness. Parents of female students had higher level of knowledge. All of the statistical results were significant (*p* < 0.05, Table 6).

## 4. Discussion

At present, data on the knowledge, attitude and practice towards free sugar is still lacking. This study is the first to report parents’ KAP about free sugar in China. Adolescent parents had a more positive attitude towards controlling adolescents’ free sugar intake, but their perception and practice of free sugar were poor in Changsha, China. Our results suggested that higher knowledge level played a positive role in parents’ buying, storing and boot behaviors related to free sugar.

We found that the knowledge of free sugar in the parents of teenagers in Changsha, Hunan, was insufficient, and they did not understand the concept of free sugar. Only 7.4% of the respondents indicated that they knew the concept of “free sugar”. Tierney et al. [27] and Wang et al. [28] showed that even well-educated residents had very limited knowledge of free sugars. At present, the self-reported awareness rate of free sugar in Chinese parents is very low, and people in other countries have low awareness of the WHO guidelines, as well. They knew the free sugar content in vegetables, cakes, milk tea, and carbonated drinks more accurately, but the perception of free sugar content in other foods and 100% fruit juice was inaccurate. The low awareness of free sugar may be related to its limited pathways. Parents may evaluate the free sugar content based on the taste of the food. The study by Dallacker et al. [29] found that parents often underestimated the amount of sugar in food or beverages. Similarly, our results showed that only 27.5% of parents could correctly recognize the free sugar content in 100% fruit juice, which was lower than the 40% recognition rate of the American population [30]. The 100% fruit juice is a beverage that contains no added sugar but contains free sugar. Drinking it in moderation can provide high-quality vitamins and minerals, but it has a similar composition of free sugar as sugary beverages. So, excessive drinking of 100% fruit juice may cause weight gain [31,32]. Excessive free sugar intake can affect adolescent growth and development; most (89.9%) parents will first consider overweight/obesity, which was consistent with the findings of Boles et al. [33] Parents were more likely to associate the harm caused by excessive sugar intake with overweight and obesity, but they ignored other health hazards, such as diabetes, nonalcoholic fatty liver, and cardiovascular disease.

A good attitude is the foundation of good behavior, and positive attitude can help improve the level of knowledge and correct inappropriate behavior. Most parents showed a very positive attitude towards controlling the intake of free sugar among adolescents in this study, and most of them agreed to reduce or control added sugar intake. An intervention study conducted by Christopher et al. [34] to reduce the intake of children’s sugary drinks showed most parents were neutral or supportive about the establishment of “SSB-free zones”. Parents’ diet is a key factor that affects their children’s diet. To a certain extent, it affects the attitude of parents in feeding children. Many researchers believe that by improving parental behavior, it could help parents to improve their attitudes toward controlling the intake of sugary foods/free sugar in adolescents [34,35,36,37]. Some family physicians and parents in the United States believe that adding too much sugar in the diet will have a negative impact on health, and recommended controlling the added sugar in food and beverages [38,39]. A positive attitude is a necessary prerequisite for behavior improvement. The awareness of Chinese residents about sugar is constantly improving, and their attitude will become better and better.

Adolescents’ autonomy in eating behavior is gradually increasing, and they can choose or buy food by themselves. This may be one of the reasons why parents’ purchase behavior was better than storage behavior. Parents’ subjective reluctance to purchase beverages and desserts for children could also reflect parents’ good purchasing behavior. However, parents’ performance of storing beverages and desserts at home was poor. Compared with storing drinks at home, more parents stored desserts at home, which may be related to their own desire for sweetness [40]. Ares et al. [41] also found that the packaging or label images will increase consumers’ willingness to buy desserts, which may cause parents to buy more desserts at home. Previous studies have shown that when children were rewarded with food or allowed to eat desserts on a regular basis, this may lead to the increment of likelihood that children will eat sweets every day [42]. In this study, 33.6% of parents used beverages as rewards, and 42.5% used dessert as rewards when children performed well. There is a strong need for this to be improved. Many parents have realized that the consumption of children’s sugary drinks is related to their own drinking behavior [12], but only about one-third of parents in this study consciously avoided drinking or buying beverages in front of their children. In this regard, it was found that the higher the level of education, the better the performance of the parents. A positive correlation between parents’ low education level and high free sugar intake of adolescents was found in the HELENA-CSS study. A cohort study in the Netherlands also found a negative correlation between the mother’s education level and children’s consumption of sugary drinks. It was shown that the higher the education level, the more likely the parents were to formulate rules for children’s sugary foods, so the less intake of children’s sugary foods [11,43,44].

Knowledge is the driving force for changing behaviors, and good knowledge can guide individuals to take corrective actions [45]. In our research we found a weak but meaningful positive correlation between knowledge level and practice through correlation analysis. Relevant knowledge guidance can help parents understand the nutritional value and sugar content of food, which is conducive to better food choice for children [46]. Therefore, it is very necessary to provide health education about free sugar to adolescents’ parents. Binary logistic regression analysis results showed that being female was a protective factor of higher levels of awareness than male parents. Women are the leader of daily life in Chinese families. They will be more involved in family eating behaviors and food choices, and pay more attention to children’s health, so women may have a higher awareness of free sugar in the population than men. Besides, the cognitive level of parents tends to increase with the increase of academic qualifications. In the HEIA study, the consumption of sugary fruit juice beverages of children decreased significantly after intervention on the nutritional knowledge of sugary fruit juice beverages of parents with a low educational background, indicating that knowledge intervention for parents helps to control children’s free sugar intake [47]. We also found that female children are protective factors in parents’ free sugar knowledge level. The current social cultural and environment makes being female pay more attention to their body shape and appearance, and parents also pay more attention to the child’s body shape, especially the attention of female parents to the body shape of their daughters [48,49]. We believe that due to the social environment and the subjective attitudes of the parents towards the girl’s body shape, it is possible to promote parents of the girl to pay more attention to the knowledge about sugar intake.

A couple of countries have taken some strategies to control residents’ sugar intake, for example, the United Kingdom, the United States, Mexico and other countries implemented sugar taxes [50,51,52], and Canada and other countries set “traffic light” warning labels on food packaging [53]. Some regions also prohibited schools from selling high-calorie food to students to improve the eating environment [54]. These kinds of measures controlled the sugar consumption of residents to some extent. In addition, the establishment of free sugar content restriction standards for prepackaged foods, and encouraging food processing companies to reduce the use of free sugar in processed foods, support enterprises to develop and produce low-sugar and sugar-free foods, and these measures may help the public reduce free sugar intake. Parents have an important influence on controlling the sugar intake of adolescents. Therefore, we can improve the behavior of parents by formulating corresponding social norms. It is also possible to expand the nutritional knowledge channels of parents through self-media and WeChat platforms. At present, many schools in our country have established WeChat groups for parents, and parents can be guided through this platform. Parents’ high participation in this study indicated parents were very concerned about their children’s health issues. Therefore, it is a feasible method to carry out nutrition classes for parents through schools. This study also provided a reference direction for future investigations. Learn more about the influence of social status, economic status, gender and other factors on parents’ KAP of free sugar, and provide reference materials for the formulation of sugar control measures in the future. However, there is a limitation. The sample representativeness was limited because we only collected data on adolescent parents in Changsha, Hunan. In addition, it is recommended to continue to combine relevant research in other regions with the results of this study.

## 5. Conclusions

Our results showed that adolescent parents in Changsha, China generally have a good attitude but moderate knowledge and practice regarding free sugar. Parents have a low awareness of the concept of free sugar, and have a poor awareness of the content of free sugar in 100% juice. We also found that only a third of parents avoided buying or drinking sugary drinks in front of their children. Parents’ gender, education level and children’s gender are the factors that affect parents’ perception of free sugar. These findings can provide useful information for health authorities to control free sugar intake in the future.

## Figures and Tables

**Figure 1 ijerph-17-04003-f001:**
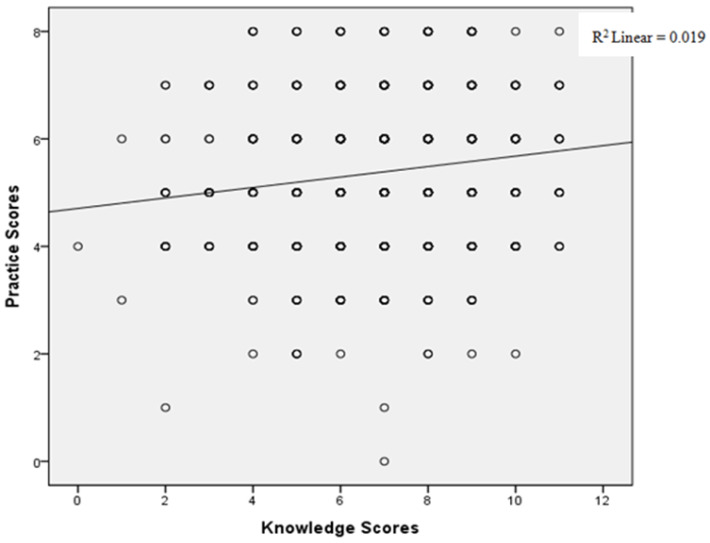
Relationship between knowledge score and practice score, based on Spearman’s correlation coefficient.

**Table 1 ijerph-17-04003-t001:** Demographic characteristics of participants (parents and adolescent, *n* = 1136).

Characteristics	Frequency	Percentage	Mean	SD
**Parents**				
Gender				
Male	336	29.6		
Female	800	70.4		
Age			41.76	5.27
Ethnicity				
Han nationality	1095	96.4		
Minority	41	3.6		
Education level				
Primary school	26	2.3		
Junior high school	280	24.6		
High school/vocational school	484	42.6		
University and above	346	30.5		
Monthly household income per capita				
≤1000 yuan	23	2.0		
1001–3000 yuan	203	17.9		
3001–5000 yuan	310	27.3		
5001–7000 yuan	231	20.3		
≥7001 yuan	369	32.5		
BMI				
Wasting	67	5.9		
Normal	768	67.6		
Overweight	256	22.5		
Obese	45	4.0		
**Adolescent**				
Gender				
Male	597	52.6		
Female	539	47.4		
Age			13.57	0.69
Only child	653	57.5		
BMI				
Wasting	45	4.0		
Normal	786	69.2		
Overweight	169	14.6		
Obese	136	12.0		
Total	1136	100		

**Table 2 ijerph-17-04003-t002:** Knowledge, attitude and practice scores and classifications among parents.

Items	Mean (P_25_, P_75_)	>P_75_ (*n* %)
Knowledge	7.04 (6, 8)	241 (21.2%)
Attitude	5.17 (5, 6)	580 (51.1%)
Practice	5.39 (4, 6)	218 (19.2%)

**Table 3 ijerph-17-04003-t003:** Parents’ perception of free sugar.

Questions	Correct Responses	
	Frequency	Percentage
Know the definition of free sugar.	84	7.4
Know the recommended daily sugar intake in The Chinese Dietary Guidelines.	386	34.0
Know recommended ratio of free sugar to total daily energy. (≤10%)	346	30.5
Knowledge of free sugar content in food.		
Biscuits, cakes	892	78.5
Vegetable, fruit	967	85.1
Sugar-sweetened beverages	944	83.1
100% fruit juice	312	27.5
Honey	692	60.9
Do you think that excessive intake of sugary drinks will affect the growth and development of adolescents? (YES)	1024	90.1
What diseases do you think excessive sugar intake may be related to?		
Dental caries	856	75.4
Overweight or obese	1021	89.9
Chronic diseases in adulthood	474	41.7

**Table 4 ijerph-17-04003-t004:** Parents’ attitude towards free sugar (*n*, %).

Items	Agree	Neutral	Disagree
Eating too much sugary food is not good.	717 (63.1)	373 (32.8)	46 (4.0)
Drinking too much sugar-sweetened beverages is addictive.	910 (80.1)	40 (3.5)	186 (16.4)
Teenagers should reduce the consumption of sugary foods.	1028 (90.5)	57 (5.0)	51 (4.5)
Drinks and desserts should limit the amount of sugar added.	1061 (93.4)	39 (3.4)	36 (3.2)
Limiting sugary foods is good for your child’s health.	1068 (94.0)	39 (3.4)	29 (2.6)
Parents should consciously control children’s intake of sugary foods.	1094 (96.3)	36 (3.2)	6 (0.6)

**Table 5 ijerph-17-04003-t005:** Buying, storing, and boot behaviors of adolescent parents.

Questions	Good behavior	
	Frequency	Percentage
You had bought three or more drinks for your child in the past month.	67	5.9
You had stored drinks at home in the past month.	417	36.7
You had bought three or more desserts for your child in the past month.	97	8.5
You had stored desserts at home in the past month.	688	60.6
You did not use drinks as a reward when your child is doing well.	1072	94.4
You did not use dessert as a reward when your child is doing well.	1025	90.2
You were conscious of avoiding buying sugar-sweetened beverages in front of your child.	394	34.7
You were conscious to avoid drinking sugar-sweetened beverages in front of your child.	422	37.1

**Table 6 ijerph-17-04003-t006:** Related factors affecting parents’ knowledge to free sugar.

Factors	b	Waldχ2	*p*	OR (95%CI)
Parents gender	0.563	10.383	0.001 ^∗∗∗^	1.756 (1.247, 2.473)
Education level of parents	0.350	13.462	0.000 ^∗∗∗^	1.419 (1.177, 1.710)
Students gender(girl)	0.290	3.864	0.049 ^∗^	1.336 (1.001, 1.783)

Notes: Logistic regression was applied in analysis; * *p* < 0.05, *** *p* < 0.001.

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
