# Peer review of "Knowledge, Attitude, and Practice of Adolescent Parents on Free Sugar and Influencing Factors about Recognition"

_ijerph, 2020, doi:10.3390/ijerph17114003_

Round 1

Reviewer 1 Report

Tang et al showed that adolescent parents’ gender, education level and children’s gender are factors that affect parents’ perception of free sugar, at least in Changsha. The finding is interesting and can be useful to control free sugar intake to benefit health of adolescents. Here are comments and suggestions.

  1. Based on the statistical analysis, the author used nonparametric tests instead of parametric tests. Is there any particular reasons? Has the author ever tried to test results of different groups with normality analysis to know if results follow Gaussian distribution?

  1. In the study design, the author mentioned two middle schools were randomly selected. How the selection was performed? Were those two schools representative enough for schools in Changsha? It will be helpful to provide information about those two schools compared to others and also methods about how those two were selected.

  1. In discussion, the author mentioned the current study is the first to report parents’ KAP about free sugar in China. Has the similar research reported before outside of China? If yes, what is the conclusion and how the results obtained here are compared with others? It will be good to include the information in the discussion if available.

  1. How students’ gender had effects on parents’ knowledge to free sugar? Is that related to the society’s expectation to girls, such that parents set higher standards?

  1. The author mentioned women in China are the leader of the families and explained the possibilities about how women influence their children. It is an interesting point. Since family education can come from both father and mother side, it will be interesting to also know whether western countries if similar study has done before would draw similar conclusions.

  1. Is the single child as a influencing factor? How about single parent? Is there any single parent case included in the study?

  1. There is no explanation about table 7. There is no table 5 and 6 too.

Author Response

  • Based on the statistical analysis, the author used nonparametric tests instead of parametric tests. Is there any particular reasons? Has the author ever tried to test results of different groups with normality analysis to know if results follow Gaussian distribution?

Thank you for this important question. Before conducting statistical analysis, we have tested that the data did not follow the normal distribution, but belonged to a skewed distribution, so we used a non-parametric test method for analysis. Line 137-140

  • In the study design, the author mentioned two middle schools were randomly selected. How the selection was performed? Were those two schools representative enough for schools in Changsha? It will be helpful to provide information about those two schools compared to others and also methods about how those two were selected.

Thank you for your valuable suggestions. The sampling method selected in this research is cluster random sampling. We selected 5 administrative districts in the urban area for investigation according to the division of administrative districts in Changsha City. Each district randomly selected 2 secondary schools, and each school randomly selected 2 classes in the first grade and the second grade. A total of 10 schools and 40 classes are included, and the selected samples are representative. See Line 83-88

  • In discussion, the author mentioned the current study is the first to report parents’ KAP about free sugar in China. Has the similar research reported before outside of China? If yes, what is the conclusion and how the results obtained here are compared with others? It will be good to include the information in the discussion if available.

Thanks for this precious recommendation. We searched the database and found that there are no similar studies domestic and overseas. But we found out about the knowledge of added sugar and WHO's guide to free sugar. During the search of literature, we have found more literature related to sugar-sweetened beverages, including the intake of sugar-sweetened beverages, the health effects of excessive intake, and the strategies taken to control the intake of sugary foods. For example:

[16] Ferrari GLM;Kovalskys I;Fisberg M;Gomez G;Rigotti A;Sanabria LYC;Garcia MCY;Torres RGP;Herrera-Cuenca M;Zimberg IZ;Guajardo V;Pratt M;Previdelli AN;Scholes S;Celis-Morales CA;Sole D. Anthropometry, dietary intake, physical activity and sitting time patterns in adolescents aged 15-17 years: an international comparison in eight Latin American countries. BMC Pediatr. 2020, 20 (1):24, doi:10.1186/s12887-020-1920-x

[25] Gui ZH;Zhu YN;Cai L;Sun FH. Sugar-Sweetened Beverage Consumption and Risks of Obesity and Hypertension in Chinese Children and Adolescents: A National Cross-Sectional Analysis. Nutrients. 2017, 9 (12), doi:10.3390/nu9121302

[26] Tierney M; Gallagher AM; Giotis ES; Pentieva K. An Online Survey on Consumer Knowledge and Understanding of Added Sugars. Nutrients. 2017, 9 (1), doi:10.3390/nu9010037

[27]Wang F; Diangelo CL; Marsden SL; Pasut L;Kitts D; Bellissimo N. Knowledge of Sugars Consumption and the WHO Sugars Guideline among Canadian Dietitians and Other Health Professionals. Can J Diet Pract Res. 2020,   :1-4, doi:10.3148/cjdpr-2020-004

  • How students’ gender had effects on parents’ knowledge to free sugar? Is that related to the society’s expectation to girls, such that parents set higher standards?

Appreciate for this question and suggestion. In the revised manuscript, we added a discussion about the free sugar knowledge level of the parents of girls, and added literature to support it.

We also found that children as girls are the protective factors of parents' free sugar knowledge level. The current social cultural and environment makes female pay more attention to their body shape and appearance, and parents also pay more attention to the child's body shape, especially the attention of female parents to the body shape of their daughters. [47, 48]. We believe that due to the social environment and the subjective attitudes of the parents towards the girl’s body shape, it is possible to promote parents of girls to pay more attention to the knowledge about sugar intake. Line 304-310

[47] Di Giacomo D;De Liso G;Ranieri J. Self body-management and thinness in youth: survey study on Italian girls. Health And Quality Of Life Outcomes. 2018, 16,

[48] Handford CM;Rapee RM;Fardouly J. The influence of maternal modeling on body image concerns and eating disturbances in preadolescent girls. Behaviour Research And Therapy. 2018, 100  :17-23,

  • The author mentioned women in China are the leader of the families and explained the possibilities about how women influence their children. It is an interesting point. Since family education can come from both father and mother side, it will be interesting to also know whether western countries if similar study has done before would draw similar conclusions.

Thanks for your comment. We consulted the literature and found that the society of western countries also has its unique inherent attributes for women, the gender gap in work is obvious, and it also has its special attributes in family life [1]. In a review by Miller et al., there were situations that the main caregivers of western families are also female[1, 2]. In the current studies, it is also found that fathers are more and more involved in the process of child-rearing. However, there are certain cultural differences between western countries and China. And it would be interesting if conduct similar research in western countries.

[1] Hesmondhalgh D, Baker S. Sex, gender and work segregation in the cultural industries. Sociol Rev. 2015;63(Suppl 1):23‐36. doi:10.1111/1467-954X.12238

[2] Miller AL, Miller SE, Clark KM. Child, Caregiver, Family, and Social-Contextual Factors to Consider when Implementing Parent-Focused Child Feeding Interventions. Curr Nutr Rep. 2018;7(4):303‐309. doi:10.1007/s13668-018-0255-9

  • Is the single child as an influencing factor? How about single parent? Is there any single parent case included in the study?

Thank you for the precious recommendation. We used "whether is an only child" as an independent variable for Binary Logistic regression analysis, and the results showed that this factor did not affect the parents' knowledge level in this study. (Line195-203)

It is a pity that when we collected demographic data, we did not investigate the family composition structure. We agree that "single parent family" is a very meaningful factor, and we hope that future research will explore this point.

  • There is no explanation about table 7. There is no table 5 and 6 too.

Thanks for your precious recommendation, we are sorry that our mistake bothered you. The correct number of “Table 7” is “Table 6”. (Line 209) We revised the form number in the newly submitted manuscript. The explanation about “Table 6” is the part of “3.3. Relevant factors affecting the knowledge of adolescent parents”. (Line 194-202) .We have carefully checked the full text to avoid similar errors, thanks again for your kindly reminder.

Reviewer 2 Report

Thank you for the opportunity to review this article. The work is interesting, but some aspects should be taken into account before publication.

Comments and suggestions for Authors:

  • I you had a bioethics committee, give us your consent number.
  • Why was the city of Changsha chosen? What distinguishes this city?
  • Why are there not many male parents in the study group? It is best if the groups are equal.

Author Response

  • I you had a bioethics committee, give us your consent number.

Thank you for your kindly reminder. The IRB number of this study is “XYGW-2019-025”. We have written the IRB number in the newly submitted manuscript. Line 93

  • Why was the city of Changsha chosen? What distinguishes this city?

Thank you for this important question. The reasons why we choose Changsha City are as follows: (1) Our school is located in this city, which brings great convenience to our investigation. (2) Changsha is an important central city in the middle reaches of the Yangtze River, with nearly 7.4 million permanent residents. It is a medium-sized city and has certain representativeness nationwide. Line 82-83

  • Why are there not many male parents in the study group? It is best if the groups are equal.

Thank you for this important question. Our sampling method is cluster sampling. In the study, our main purpose was to investigate the situation of KAP related to free sugar of the primary caregivers of middle school students. Therefore, in the data we collected, the gender ratio of male and female was not equal, which also reflects that female in Chinese families occupy a certain dominant position in the process of life and child-rearing. Line 89--91

Reviewer 3 Report

The aim of the cross-sectional study is to evaluate the adolescent parents' knowledge, attitude and practice towards free sugar among 10 middle schools in China. The study was conducted correctly. Some parts of the paper can be improved before publication. Some notions of nutrition should be included in the introduction, in particular the difference between simple sugars and complex carbohydrates and their different influence on health should be highlighted. The discussion should be improved because at the moment it is quite confusing. Making it structured would probably improve the interpretation of the data. At the moment the same concept is repeated several times about the need to improve parents' knowledge about simple sugars and the application of these rules in daily practice. Other aspects such as gender differences, the influence of education and the social class could be further investigated. It would also be useful to highlight what practical suggestions can be given to transfer nutritional knowledge into the diet. 

Author Response

  • Some notions of nutrition should be included in the introduction, in particular the difference between simple sugars and complex carbohydrates and their different influence on health should be highlighted.

Thanks for your precious recommendation. Free sugar includes monosaccharides and disaccharides added to foods and beverages by manufacturers, cook or consumers, besides naturally presenting in honey, syrups, fruit juices and fruit juice concentrates. Nevertheless, the endogenous sugars in whole fresh fruits and vegetables don’t belong to it. Free sugars are simple carbohydrates, and studies have shown that carbohydrates are the main source of energy for the body. Contrary to complex carbohydrates derived from intact and unprocessed vegetables, excessive amounts of simple carbohydrates can easily lead to health problems such as weight gain, while complex carbohydrates can prevent diet-related chronic non-communicable diseases[3].(Line 34-38)

[3] Ferretti F;Mariani M. Simple vs. Complex Carbohydrate Dietary Patterns and the Global Overweight and Obesity Pandemic. Int J Environ Res Public Health. 2017, 14 (10), doi:10.3390/ijerph14101174

  • The discussion should be improved because at the moment it is quite confusing. Making it structured would probably improve the interpretation of the data.

Thank you so much for the comments. We revised the structure of the discussion section in the new manuscript. (Line 218-221, 225-228, 240, 241-242, 243-255, 290-310) The general framework is as following:

At first, we gave a general overview of the findings of this study and showed the innovation of this study.

In the second paragraph, we described the current situation of the awareness of free sugar and existing problems among parents of adolescents in Changsha City. Then we discussed the parents' recognition of the content of free sugar in food, especially the situation of 100% juice, and describe the parents' perception of the health hazards of excess free sugar.

Then we showed the attitude of parents in this study to control the intake of free sugar in adolescents, and the attitude of parents found in the study of Christopher et al. on sugar-sweetened beverages. And citing literature to support positive attitudes can help improve parental behavior.

In the behavior section, we explained the current status and possible reasons of parents' purchase and storage behavior in this study through literature support, and discussed the problems and related factors in parental guidance behavior.

Next, we discussed the relevant factors of the free sugar cognition level found in the Binary Logistic analysis of this study. Three factors were discussed separately: parental gender, parental education level, and child's gender.

Finally, we retrieved the strategies taken abroad to control free sugar intake, and made several suggestions based on the results of this study.

Thank you again for your suggestion, we think that the discussion is clearer after this modification.

  • At the moment the same concept is repeated several times about the need to improve parents' knowledge about simple sugars and the application of these rules in daily practice. Other aspects such as gender differences, the influence of education and the social class could be further investigated. It would also be useful to highlight what practical suggestions can be given to transfer nutritional knowledge into the diet.

Thank you for your precious suggestion. We have modified the same concept in the discussion section, simplified the expression. (Line224-225, 240-2242, 294-295)

 In addition, the suggestions in the penultimate paragraph of the discussion section were completely revised. As following:

Parents have an important influence on controlling the sugar intake of adolescents. Therefore, we can improve the behavior of parents by formulating corresponding social norms. It is also possible to expand the nutritional knowledge channels of parents through self-media and WeChat platforms. At present, many schools in our country have established WeChat groups for parents, and parents can be guided through this platform. Parents’ high participation in this study indicated parents were very concerned about their children’s health issues. Therefore, it is a feasible method to carry out nutrition classes for parents through schools. This study also provided a reference direction for future investigations. Learn more about the influence of social status, economic status, gender and other factors on parents' KAP of free sugar, and provide reference materials for the formulation of sugar control measures in the future. (Line 316-325)

Round 2

Reviewer 2 Report

I have no comments.

Author Response

Thanks so much for your valuable comments and suggestions on our manuscript.

Reviewer 3 Report

The authors have followed my directions and the paper seems to me to be much clearer and more readable.

The discussion can be further improved. Some considerations are missing about possible policies that could be useful to reduce sugar consumption among young people. For example the sugar tax or the reduction of targeted advertising. What do the authors think should be done to counteract this issue?

Author Response

The discussion can be further improved. Some considerations are missing about possible policies that could be useful to reduce sugar consumption among young people. For example the sugar tax or the reduction of targeted advertising. What do the authors think should be done to counteract this issue?

Thanks for your precious suggestion. Measures such as sugar taxes and traffic lights have certain beneficial effects on controlling the intake of free sugar. In addition, the establishment of free sugar content restriction standards for pre-packaged foods, and encourage food processing companies to reduce the use of free sugar in processed foods, support enterprises to develop and produce low-sugar and sugar-free foods, these measures may help the public reduce free Sugar intake. We can guide parents to make reasonable choices of food through pre-packaged food labeling while educating parents. (Accept the revised version: Line 296-299 OR Previously submitted version: Line 315-318)